# A Systematic Review of Blockchain Technology Adoption Barriers and Enablers for Smart and Sustainable Agriculture

Gopi Krishna Akella [1,*], Santoso Wibowo [1], Srimannarayana Grandhi [1] and Sameera Mubarak [2]

1   School of Engineering & Technology, Central Queensland University, Melbourne, VIC 3000, Australia; s.wibowo1@cqu.edu.au (S.W.); s.grandhi@cqu.edu.au (S.G.)
2   UniSA STEM, University of South Australia, Adelaide, SA 5001, Australia
*   Correspondence: g.akella@cqu.edu.au

**Abstract:** Smart and sustainable agricultural practices are more complex than other industries as the production depends on many pre- and post-harvesting factors which are difficult to predict and control. Previous studies have shown that technologies such as blockchain along with sustainable practices can achieve smart and sustainable agriculture. These studies state that there is a need for a reliable and trustworthy environment among the intermediaries throughout the agrifood supply chain to achieve sustainability. However, there are limited studies on blockchain technology adoption for smart and sustainable agriculture. Therefore, this systematic review uses the PRISMA technique to explore the barriers and enablers of blockchain adoption for smart and sustainable agriculture. Data was collected using exhaustive selection criteria and filters to evaluate the barriers and enablers of blockchain technology for smart and sustainable agriculture. The results provide on the one hand adoption enablers such as stakeholder collaboration, enhance customer trust, and democratization, and, on the other hand, barriers such as lack of global standards, industry level best practices and policies for blockchain adoption in the agrifood sector. The outcome of this review highlights the adoption barriers over enablers of blockchain technology for smart and sustainable agriculture. Furthermore, several recommendations and implications are presented for addressing knowledge gaps for successful implementation.

**Keywords:** blockchain technology; smart and sustainable agriculture; agribusiness; decentralized applications; barriers; enablers





## 1. Introduction

A blockchain is an immutable record of transactional data, stored in encrypted form on a distributed network of mutually anonymous nodes [1]. Over the last few years, blockchain technology has received much attention due to its numerous benefits for the agriculture sector. This is due to its characteristics of immutable, decentralized, unanimous, and enhanced security [2,3].

Smart and sustainable agriculture relies on the availability of reliable data for informed decision making by using expert systems and Internet of Things (IoT) devices. The existing e-agriculture systems provide information to farmers, producers, farming communities, and suppliers to make appropriately informed decisions [3]. These systems significantly improve productivity by accessing reliable data from food producers and farmers [4]. However, these systems have limitations such as a lack of flexibility, collect insufficient information, and provide biased information [5]. These systems are not appropriate since the agriculture industry worldwide is distributed, and multi-actor based, with different stakeholders, such as farmers, wholesalers and retailers, suppliers, distributors, financial institutions, distributors, and consumers. Many research studies suggest that blockchain technology is an ideal solution for solving the problems of these e-agriculture systems [6].

This is due to the key characteristics of blockchain such as reliability, traceability, proof-of-ownership, proof-of-work, and encrypted and immutable data which provide trustworthy environments with minimum or no intermediaries [7].

Smart and sustainable agriculture refers to a set of solutions that increases agricultural efficiency and reduces the ecological footprints of farming [8]. It involves a variety of systems and technologies such as the Internet of Things (IoT) and information communication technology (ICT). There are many processes involved from production on the farm, to transformation, distribution, and trading to reach the consumers [2]. The food supply chain is a complex long process having multiple intermediaries with an enormous number of transactions from farm to fork. Products and services flow in one direction and financial exchanges happen in the opposite direction. Information is considered a great asset these days and flows in both directions in agrifood supply chains [9]. However, the asymmetric nature of information and misinformation in a supply chain transaction is a constant challenge. Moreover, customers seek information about the origin of the food they consume. They need to verify their products for sustainability and require a system to know the product information [7]. Thus, there is huge pressure on food suppliers to maintain a sustainable system that manages the friction between the demand and supply of agrifood products, and food provenance as well as eliminating centralized authority.

Prior studies [10–16] have been conducted on understanding the effect of blockchain on smart and sustainable agriculture. These studies showed that blockchain technology can provide a solution to the issues relating to double spending, information asymmetry, traceability, and data security in the agriculture sector. However, there is a limited study on the successful adoption and implementation of blockchain technology for achieving smart and sustainable agriculture practices. Furthermore, it does not discuss the full-stack development of blockchain applications for all agricultural processes. Therefore, it is important to investigate the main issues concerning blockchain adoption for achieving smart and sustainable agriculture. This study aims to present the blockchain adoption enablers and barriers and necessary actions to manage them for smart agriculture and sustainable agricultural practices. Thus, the following research question is formulated as follows: *What are the barriers and enablers in blockchain technology adoption for smart and sustainable agriculture?*

To answer this research question, we rely on current literature to constitute a background of enablers and barriers to blockchain technology adoption in agriculture. To achieve this objective the current research adopts a qualitative systematic literature review approach using PRISMA (Preferred Reporting Items for Systematic Reviews and Meta-Analysis) technique to focus on investigating the adoption status so that we can present the barriers and enablers of blockchain technology adoption for smart and sustainable agriculture. This study performs an intensive search to collect data for an in-depth thematic analysis of enablers and barriers of blockchain adoption.

The main contribution of this study is to examine the barriers and enablers of blockchain technology adoption for smart and sustainable agriculture. Based on the review, the results show that there are several barriers that need to be considered for blockchain technology adoption in smart and sustainable agriculture such as: (a) lack of government regulations, (b) resource capital requirements, (c) security and privacy concerns, (d) lack of standards, (e) trust, (f) scalability issues, (g) awareness and (h) ease of use. In addition, various enablers can contribute to the successful adoption of blockchain technology in smart and sustainable agriculture including (a) stakeholder collaboration, (b) enhancing shared responsibilities of partners, (c) increasing customer trust, quality of service, sustainable value chains and infrastructure, (d) data security and useability, (e) improving efficiency in supply chains and (f) enhancing agricultural democratization. This study offers meaningful insights to researchers and practitioners for a better understanding of the issues relating to blockchain technology adoption and provides guidance in implementing this technology for smart and sustainable agriculture.

This paper is divided into the following sections. Section 2 discusses blockchain for agriculture. Section 3 discusses the research method conducted for this study. Section 4 discusses the results of the review process. Section 5 provides a discussion and finally, Section 6 presents the conclusion of the study.

## 2. Literature Review

### 2.1. Blockchain Technology

Blockchain is a distributed ledger of transactions that stores encrypted information into a chain of blocks [17–19]. It is not possible to tamper with information as the blockchain intends to timestamp the information. The purpose of blockchain is to mitigate third parties like banks and brokers in a transaction. Since blockchain uses distributed ledger technology, it is resilient, secure, reliable, decentralized, and fraud-proof. From a computing paradigm, blockchain is a decentralized shared database system. From a technical point of view, it is a public distributed ledger system; and from a business perspective, it is a huge network of transactions of data and assets without any intermediaries [18].

Agriculture supply chains are challenging because there is a lack of transparency, traceability, and trust. Moreover, food provenance, quality, and safety have become major concerns for consumers in recent years due to food epidemic incidents. Even though various web-based e-agriculture systems are available and in use, these systems are not transparent and reliable as they are centralized, asymmetric, and incompatible [9]. Moreover, the existing systems lack consumer trust due to the unavailability of easy-to-retrieve reliable information about the provenance of food products [17]. Thus, blockchain technology is a useful tool for providing an effective solution to the above-mentioned issues.

Blockchain technology integrates many existing technologies for capturing data, storage, and dissemination [18]. The fundamental characteristics of blockchain technology are (a) distributed and decentralized where the nodes in the network are completely distributed and there is no owner to manage the network (b) traceability and tamper-proof in which the algorithms used in blockchain ensures that the data has consistency and integrity (c) autonomous and peer-to-peer whereby the nodes in the network are homologous as the blockchain network is essentially a peer-to-peer network, i.e., the nodes are not bound to the network and (d) automatic execution of programs where the financial transaction, records, and queries of data are executed based on pre-written programs called as smart contracts [20].

Blockchain networks are usually open to all parties, where anyone can join and see transactions that will happen on the networks. However, these blockchains could be permissioned or permission-less i.e., public, or private, providing security and authentication. Blockchain allows a sharable ledger of transactions to be read, validated, and stored in a chain of blocks [21]. Systems on blockchain work in a distributed manner where independent multiple agents, or participants (as nodes) connect to the network by using P2P communications. Thus, resulting in a decentralized ecosystem with no central authority [14].

E-agriculture systems have been studied for efficient and effective farm management as they provide access to useful information for informed decision making which leads to increased productivity [6,15,16]. However, these e-agriculture systems have been proven inefficient due to limitations such as human bias, lack of trust and transparency in transactions, and asymmetric information during supply chain activities. Smart agriculture systems comprise sensor technology such as GPS and IoT enabled, cloud computing and mobile technologies which have higher efficiency as compared to the e-agriculture system [22]. Remote monitoring, field inspection, soil conditions, water and waste management, and livestock monitoring can be managed by intelligent systems. Data collection, data integrity and security have been the major concerns for these e-agriculture systems. Moreover, hackers can easily intercept the IoT-enabled farm management systems and tamper with the data. The salient features of blockchain technology could solve the operational problems of e-agriculture systems for smart agriculture [10].

## 2.2. Application of Blockchain Technology for Smart and Sustainable Agriculture

Blockchain is an immutable distributed ledger implemented without a central authority [23]. The transactions stored on a blockchain cannot be altered or removed as every node (user) connected with the blockchain network has a copy of all transactions. Thus, the transaction and authenticity can be publicly verified. Blockchain is a distributed knowledge system, with key components such as consensus algorithm, smart contracts, transparency, and proof-of-ownership, which can be a reliable infrastructure for the sustainable development of smart agriculture applications [24].

Farmers use smart agriculture systems to increase production, optimize input use, labor savings, improve market access, and timely decision-making [25]. They use smart agriculture systems for record-keeping, information gathering, managing production, and operating equipment. Over 80% of farms use these systems on grain farms, and GPS equipment, and 26% for training and education [25]. Some farmers rely on online free software tools such as the Bureau of Meteorology and other weather applications. A few of them utilize the free software from private organizations such as AgWorld, and GrainCorp [26]. The agriculture systems have shown significant improvements in agricultural productivity and food supply chains [4]. Agriculture 4.0 is taken into consideration by many researchers and organizations, which includes precision agriculture, blockchain, IoT and big data for all aspects of sustainable agriculture. Precision agriculture is a farm management system that focuses on quantifying the crop or livestock production by monitoring, measuring, and responding to field metrics for low-input, highly productive sustainable agriculture [27]. These systems enable farmers to visualize and analyze the data for healthy crop growth and maintain sustainable agricultural practices.

Information asymmetry, security, proof of work (PoW), traceability and provenance are major issues in existing e-agriculture systems [18,28–32]. The adoption of blockchain technology is envisioned to be useful for smart agriculture by introducing new dimensions. This review conducts thorough research in finding the adoption enablers and barriers of blockchain technology for smart and sustainable agriculture. Kim et al. [29] suggest that agriculture supply chains need more studies towards managing trust among stakeholders using blockchain technology. Thejaswini et al. [30] propose decentralized applications as a solution for issues in agriculture. Vangala et al. [31] also suggest that smart contracts and blockchain technology can support smart agriculture. Awan et al. [28] state that IoT and blockchain technology integrated applications will be available by 2030 for smart agriculture. They argue that IoT as well as blockchain adoption in agriculture is still in the early stages. Zhang [18] argues that blockchain could simplify the credit checking and financing process using blockchain technology in agriculture. Zhu et al. [32] insist that a secure data-sharing platform is needed by using big data and blockchain technology. The literature review identifies that there is significant research done on incorporating blockchain in the agriculture industry. However, the implementation of comprehensive smart agriculture systems and sustainable agricultural practices is very limited due to technical barriers. Furthermore, the literature reveals that the barriers are generic in nature in adopting any new technology and not enough emphasis is given to encompass the practical application of blockchain for smart agriculture and sustainable practices.

In the farm-to-consumer scenario, there exist complex processes, knowledge gaps and inconsistent information among the parties that lead to trust and reliability issues [18,28]. The current research presents evidence to address the capabilities of blockchain technology for agricultural supply chains and offer solutions such as provenance, secure payments, and traceability systems. Moreover, the studies identify a few security issues and seek global standards and industry-level best practices. We identify that there is a lack of distinctive studies in finding the adoption level, barriers, and enablers of blockchain technology for smart and sustainable agriculture. Therefore, this review will address the gaps such as identifying the adoption levels, enablers, and barriers so that the viability of blockchain technology is thoroughly understood and further insights can be made. The systematic

review of the literature unveils that there are significant barriers to adoption enablers to adopting blockchain technology for smart and sustainable agriculture.

## 3. Research Method

Qualitative research is a systematic inquiry into a phenomenon that includes how individuals and groups behave, how organizations function, and how interactions shape relationships in a natural setting [33]. In general, qualitative research includes data in the form of text, unlike numbers in quantitative research. Sometimes it is not possible to answer the research question just by using quantitative methods such as surveys. Busetto et al. [34] point out that numerical data collection is useful for measuring a pattern, for example, a consumer's shopping behavior. However, qualitative methods are more suited for discovering reasons such as patterns, especially hidden ones.

The qualitative research method is chosen for this study due to its flexibility, openness, and responsiveness to the context. A qualitative study can explore the topics of research that have not been studied or understood thoroughly. For a thorough qualitative study, an in-depth review of existing literature helps the researcher to identify relevant issues about the research topic of inquiry [35].

### 3.1. Criteria of Selection

This systematic literature review includes peer-reviewed articles published in journals. It is important to make well-informed, strategic decisions that are in line with the research questions in a study. For transparency and trustworthiness, all insights, knowledge, and findings are well-documented. This has been accomplished with the PRISMA technique [36].

For this systematic review, we have an explicit definition of the inclusion and exclusion criteria of the literature. This criterion is then applied to all sources collected and presented in the review. A systematic evaluation of the quality of studies is also included in the review. During this process, the sources excluded are identified and justified for exclusion. By using explicit and systematic methods for reviewing the articles and collected evidence, bias can be minimized, thus providing reliable findings from which decisions can be made [36]. A qualitative synthesis of results and meta-analysis is performed to achieve findings suitable for drawing conclusions [37].

This research study is based on the selection of articles published in databases such as ProQuest, IEEE, Web of Science, and ScienceDirect. These databases are selected because they include high-quality journals and original published articles that are relevant to the research topic. The literature search has been limited to peer-reviewed journals, as there is a plethora of literature on blockchain technology. The following criteria are considered for selecting articles from databases:

- Literature search for existing blockchain-based applications on smart agriculture and sustainable practices.
- Literature search for specific articles on blockchain technology adoption for smart and sustainable agriculture.
- Literature search for specific articles on suggested blockchain architectures and models for smart and sustainable agriculture.
- Literature search for specific articles on the benefits and barriers of applicability of blockchain for smart and sustainable agriculture.
- Keywords search based on "blockchain", "smart agriculture", "enablers", "barriers", "food supply chain", "e-agriculture systems", "traceability", "provenance", "trust", "security", and "transparency".

### 3.2. Data Collection

PRISMA is an evidence-based reporting mechanism in systematic reviews. The data collection for this systematic review is conducted by using the PRISMA flow diagram as shown in Figure 1. The search is performed between the years January 2015 and

August 2022 due to the availability of articles on blockchain technology for agriculture. Initially, 223 articles were eligible for a review from a huge collection of 2724 articles from various databases. Around 197 articles have been removed due to reasons, outcomes, non-blockchain and not smart agriculture criteria. Finally, 26 articles have been chosen for this study. After the review of the relevant articles, a thorough content analysis has been done. Figure 1 presents the process of data selection, inclusion, and exclusion of articles based on preferred eligibility criteria.

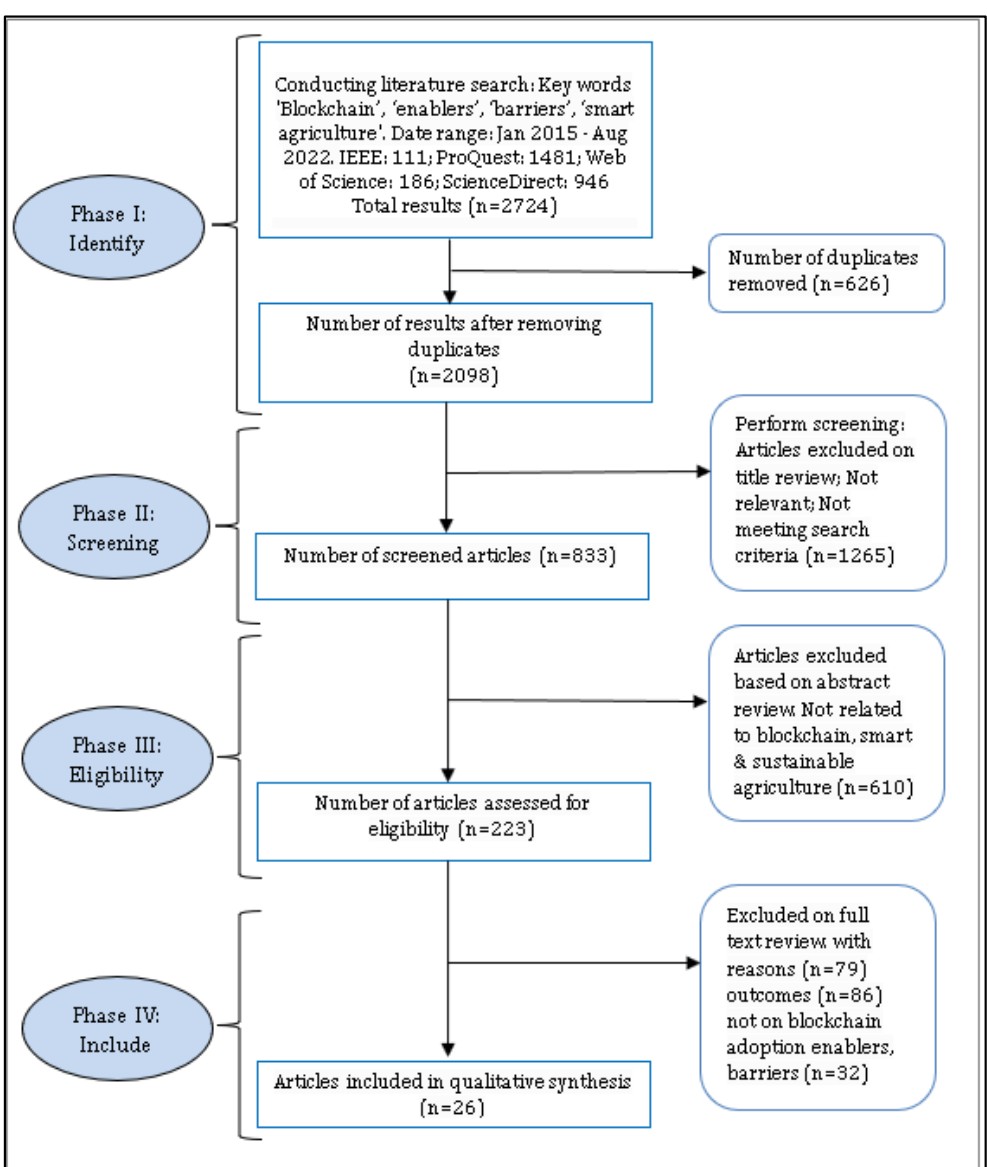

**Figure 1.** PRISMA flow diagram for systematic review.

### 3.3. Criteria for Inclusion/Exclusion

The first priority in this systematic review is to collect literature that has attributes that will make it possible to accomplish the purpose of the research study. A thorough inclusion/exclusion criteria set the boundaries for our review and the base to achieve our goal. Defining a clear criterion increases the likelihood of producing reliable and reproducible results, minimizes bias, and guards against drawing feeble conclusions. In order to achieve the purpose of this research, different factors are used for setting up the criteria. Table 1 summarizes the criteria for the selection of articles based on several characteristics such as time frame, study topic, language, and reported outcomes.

**Table 1.** Inclusion/Exclusion criteria.

| Parameters | Inclusion | Exclusion |
|---|---|---|
| Date range (Jan 2015–Aug 2022) | Articles within this date range | All articles not in the date range |
| Study topic | Blockchain for smart and sustainable agriculture | Non-blockchain and non-smart and sustainable agriculture |
| Keywords | "blockchain" AND "smart agriculture" OR "food supply chain" OR "e-agriculture systems" AND "enablers" OR "barriers" OR "traceability" OR "provenance", "trust" OR "security" OR "transparency" | Generic search words lead to biased results and confusion |
| Duplicate results | Store articles in a repository to inspect and manage unique papers | Scrutinize for redundant articles based on authors and title |
| Eligibility | Relevance to smart and sustainable agriculture | Review the abstract and exclude |
| Language | English only | Identify language and exclude |
| Selection | Conduct thorough full-text review based on keywords and outcomes matching with purpose and seem meaningful | Full-text review of papers with keywords. only on blockchain adoption enablers, barriers, and applications |

## 4. Results

The results are outlined based on the parameters such as publications on various databases and the chronological evolution of publications from the year 2015. ScienceDirect has published 11 papers that are relevant to blockchain technology for smart agriculture, and this is followed by IEEE with nine papers, Web of Science with three papers and ProQuest with three papers. A significant increase can be seen from 2019 in the number of articles published on blockchain technology for the agriculture sector. Figure 2 shows the distribution of articles on blockchain and smart agriculture between 2015–2022 in various databases.

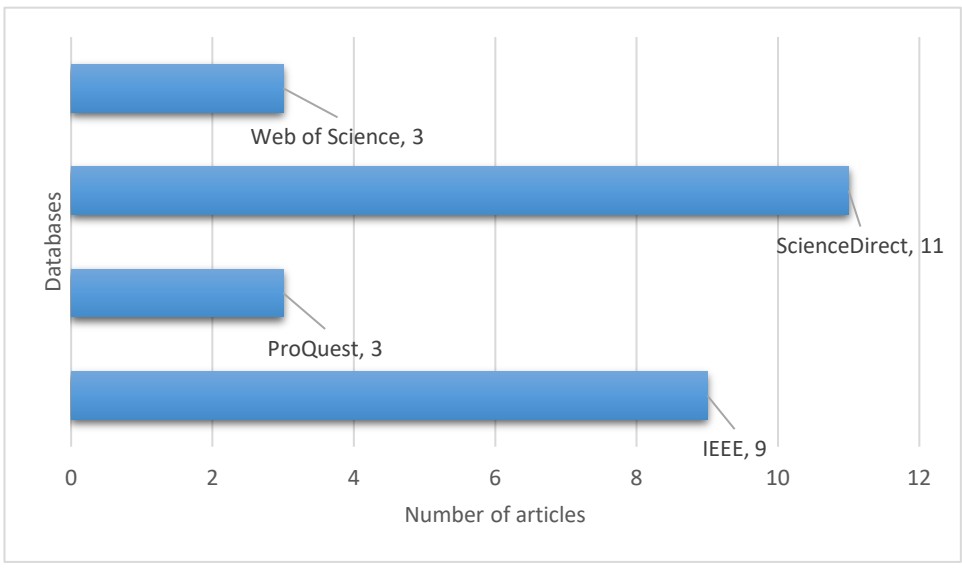

**Figure 2.** Number of articles on blockchain and smart agriculture between 2015–2022.

Although significant discussion and literature were available on blockchain technology architectures and frameworks from 2015–2019, a trend to adopt this disruptive technology to agricultural supply chains, payments, sustainability, and security can be observed in 2021. In addition, there is a clear indication that the trend is focused on implementing smart agriculture and sustainable agricultural practices from 2021 as shown in Figure 3.

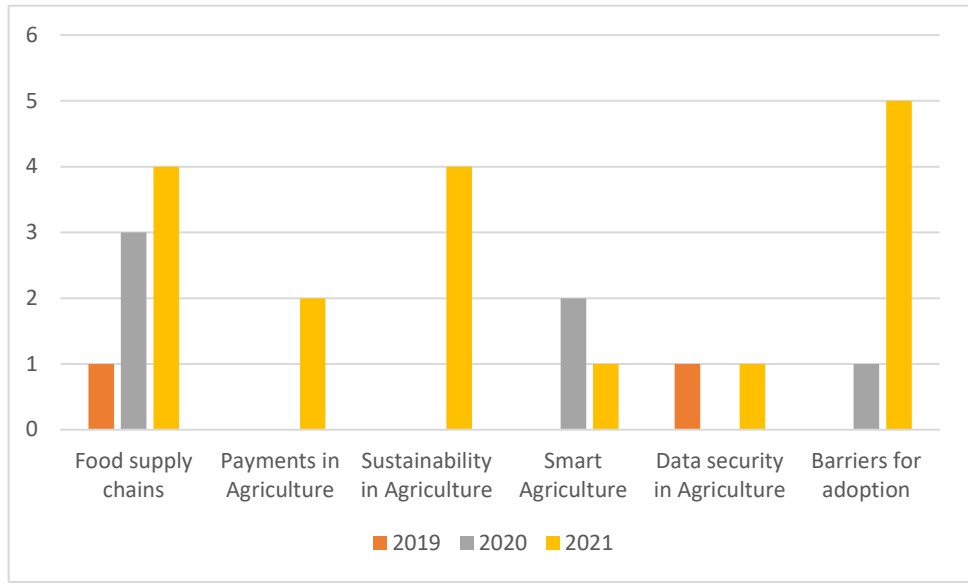

**Figure 3.** Number of articles published based on themes during 2019–2021.

An in-depth content analysis of the relevant articles is performed to identify the themes and keywords in each article. The aim is to develop a qualitative classification and categorization of the articles in this study. We have conducted a thematic analysis of the data from twenty-six articles collected to derive the outcomes of the study. We have followed the steps as (a) extract data, (b) code data, (c) categorize data, and (d) translate into themes. In this analysis, five themes are identified as adoption enablers and labeled as (1) blockchain for food supply chains, (2) blockchain for payments in agriculture, (3) blockchain for sustainability, (4) blockchain for smart agriculture, (5) blockchain for data security and one theme as a barrier and labeled as (6) barriers of blockchain adoption. The content of each article is assessed by going through the full text wherever it is essential to determine the themes. This has been an iterative process to determine all themes. Table 2 provides codes and respective themes for collected data. This has contributed significantly to the reliability of this study.

**Table 2.** Thematic analysis process.

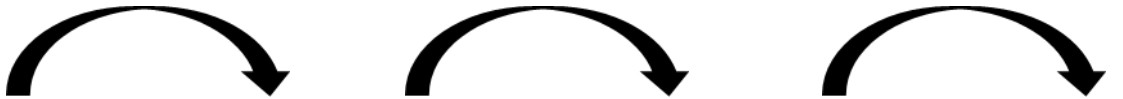

| Step 1. Extract Data | Step 2. Code Data | Step 3. Categorize Data | Step 4. Translate into Themes |
|---|---|---|---|
| Transparency in food supply chains | Transparency | Transparency | |
| Traceability using smart contracts | Traceability | | |
| Traceability for consumer trust | Traceability | | |
| Food traceability system from governmental, corporate, and consumer perspectives | Traceability | Traceability | Blockchain for food supply chains |
| Blockchain-based traceability system that ensures food safety measures | Traceability | | |
| Food traceability technology towards organic food products. | Traceability | | |

**Table 2.** *Cont.*

| Step 1. Extract Data | Step 2. Code Data | Step 3. Categorize Data | Step 4. Translate into Themes |
|---|---|---|---|
| Precision agriculture through blockchain and IoT | Blockchain application | Blockchain application | |
| A blockchain-based digital FarMarket ecosystem using smart contracts | Blockchain application | | |
| A consortium blockchain for agricultural supply chains | Blockchain application on Supply chains | | |
| Blockchain-based business financing system for the agriculture industry | Agriculture payments | Financial | Blockchain for payments in agriculture |
| Blockchain-based fast payment systems for farmers using smart contracts | Agriculture payments | | |
| Blockchain technology for sustainable performance in wine chain. | Sustainability | Sustainable agricultural practices | Blockchain for sustainability |
| Blockchain-based double chain architecture for better sustainability in agriculture | Sustainability | | |
| Blockchain architecture for sustainable e-agriculture and data management | Sustainability | | |
| Blockchain Architecture for sustainable agri-food supply chains | Sustainability | | |
| Blockchain-based trust management for agricultural green supply. | Trust management application | Smart agriculture/farm management | Blockchain for smart agriculture |
| Blockchain technology and e-agriculture systems for agricultural democratization. | Smart agriculture application | | |
| A blockchain-based fish farm platform for data integrity | Smart agriculture application | | |
| An intelligent agriculture network security system based on private blockchains | Security | Data security application | Blockchain for data security |
| Blockchain and IoT-based secure and energy efficient scheme for precision agriculture | Security | | |
| Blockchain technology adoption barriers in the agricultural supply chain | Supply chain barriers | Supply chain barriers | Barriers of blockchain adoption |
| A thematic analysis of processes and challenges of adopting blockchain for food supply chains | Challenges in supply chains | | |
| Interpretive structural modelling of barriers for blockchain in supply chains | Supply chain barriers | | |
| A review of challenges in blockchain-based food supply chains | Supply chain barriers | | |
| Possibilities and limitations of a blockchain-based solution for agro-industry | Technological limitations | Technological barriers | |
| Learning from early adopters of blockchain technology | Technological limitations | | |

Based on the systematic literature review, this study finds twenty articles discussing blockchain technology adoption enablers for smart and sustainable agriculture and six articles on the barriers of adoption. Each paper is thoroughly scrutinized for selection using our search criteria containing appropriate terms. The enablers and barriers are then systematically labeled for in-depth analysis and able to draw conclusions. Therefore, the adoption enablers and barriers are more specific to smart and sustainable agricultural practices as mentioned in Table 3 and are not common issues and terms mentioned in the literature related to the agriculture sector.

**Table 3.** List of adoption enablers and barriers from thematic analysis.

| Code | Adoption Enablers | Code | Adoption Barriers |
|------|-------------------|------|-------------------|
| AE1 | Stakeholder collaboration | AB1 | Lack of government regulations |
| AE2 | Enhancing ICT infrastructure | AB2 | Security and privacy concerns |
| AE3 | Enhancing shared responsibilities of partners | AB3 | Legal and ethical barriers |
| AE4 | Integrating with other technologies | AB4 | Lack of global standards |
| AE5 | Enhancing customer value | AB5 | Resource capital requirements |
| AE6 | Enhancing quality of service | AB6 | Lack of trust among stakeholders |
| AE7 | Enhancing trust among stakeholders | AB7 | Scalability issues |
| AE8 | Enhancing audit trails | AB8 | Ease of use |
| AE9 | Enhancing customer trust | | |
| AE10 | Enhancing fast payments | | |
| AE11 | Infrastructure development | | |
| AE12 | Improving sustainable supply chains | | |
| AE13 | Improving agricultural value chains | | |
| AE14 | Improving efficiency in supply chains | | |
| AE15 | Enhancing agricultural democratization | | |
| AE16 | Enhancing efficiency and useability of data | | |
| AE17 | Enhancing data security | | |
| AE18 | Increasing product quality via traceability | | |

Table 4 shows that there is significant adoption and implementation of blockchain technology in several areas of agriculture, which explains the extent of the adoption of blockchain technology in agriculture. It presents 18 adoption enablers of blockchain technology for smart and sustainable agricultural practices. There is adequate evidence for the adoption of blockchain, such as the platform developed by Astill et al. [38] to manage transparency in agrifood supply chains and the food traceability system by Casino et al. [39]. These decentralized blockchain systems justify the adoption in the agriculture sector by improving the infrastructure and greater degree of stakeholder collaboration which are drawbacks of centralized e-agriculture systems. A decentralized blockchain application by Cao et al. [40] and a blockchain model of precision agriculture developed by Torky et al. [41] attest to the enablers, such as enhancing customer trust and shared responsibilities of partners.

An AI-based traceability system for food supply chains developed by Qian et al. [42] is an exemplar of blockchain integration with other technologies. Consortium blockchain system [43], farm management system [44], and food traceability system [45] are a few more cases that prove blockchain adoption in the agriculture sector is plausible. Blockchain enhances fast and secure payments in the agriculture sector [46,47]. The platform developed by Rijanto [46] and the permissioned blockchain model by Pranto et al. [47] support the enhancing fast payments enabler.

The process of issuing a sustainability certification is cumbersome because it involves many channels and stages in food supply chains. However, blockchain can resolve the inconsistencies in the process of certification in the wine industry [11] thus improving sustainable supply chains. An architecture for agriculture sustainability [10], a supply chain management information system [12], and a double chain structure for better sustainability [13] are more exemplars of sustainability adoption enablers.

The blockchain-based trust management system for agriculture by Bai et al. [48] is another instance that supports enhancing trust among stakeholders. A democratization system [49] by integrating with e-agriculture systems proves that agricultural democratization is possible with blockchain technology. The permission-based blockchain fish farm management system by Hang et al. [50] showcases that blockchain enhances data efficiency and useability. Transactional data is stored in encrypted form on a blockchain system. The blockchain-based security system developed by Wu et al. [51] showcases the security enhancement of sensor device data. Another instance of a data security enabler is the blockchain-based framework in precision agriculture [52]. Therefore, in summary, we can consider these enablers as an attestation of blockchain adoption for smart and sustainable agriculture despite some technical, industrial, and organizational issues.

Table 5 provides key findings of the review of the barriers to blockchain adoption. There are several barriers in relation to the application of blockchain technology which are broadly classified into eight types. Yadav et al. [53] and Kochupillai et al. [54] find that lack of government regulations, security, privacy concerns and trust among stakeholders are barriers in agricultural supply chains. Chen et al. [55] claim that blockchain technology has integration issues and needs a common and open data standard. Etemadi et al. [56] discuss the importance of technology maturity, clear regulatory provisions, scalability and bandwidth issues, and smart contract issues as the main barriers to agricultural supply chains. Nurgazina et al. [57] point out the challenges associated with scalability, security, and privacy in the slow adoption of blockchain technology. Dede et al. [58] highlight the barriers to blockchain adoption including the uncertain regulatory environment, scalability, and technologically and socially understandable blockchain systems.

**Table 4.** Enablers of blockchain technology adoption.

| Themes | Author (s) | Enablers | Key Findings |
|---|---|---|---|
| Blockchain for food supply chains | Astill et al. [38] | Stakeholder collaboration Infrastructure development | Development of a blockchain-based platform for managing transparency in agrifood supply chains by improving stakeholder collaboration and blockchain-based infrastructure |
| | Casino et al. [39] | Enhancing ICT infrastructure | Development of a blockchain-based food supply chain application for traceability by upgrading ICT infrastructure |
| | Cao et al. [40] | Enhancing shared responsibilities of partners Enhancing customer trust | Development of an application called 'BeefLedger' for trust and traceability in food supply chains to enhance customer trust and maintain shared responsibilities of stakeholders |
| | Torky et al. [41] | Integrating with other technologies | Development of a blockchain model for precision agriculture by integrating with IoT |
| | Qian et al. [42] | Enhancing customer value Integrating with other technologies | Development of AI-based traceability systems for improving overall traceability across food supply chains to enhance customer value. A system that integrates blockchain with AI and IoT |
| | Leduc et al. [59] | Enhancing quality of service | Development of an application for farmers to collect and publish agricultural assets to enhance trust and quality of service |
| | Kyzy et al. [43] | Enhancing trust among stakeholders | Development of a consortium blockchain system for agriculture supply chains that enhances trust among all stakeholders |
| | Iftekhar et al. [44] | Enhancing audit trails Increasing product quality via traceability | Development of a blockchain-based architecture for farm management and supply chain traceability that leads to increased product quality and auditability |
| | Lin et al. [45] | Enhancing customer trust | A blockchain-based food traceability system to enhance customer trust |
| Blockchain for payments in agriculture | Rijanto [46] | Enhancing fast payments | Development of a blockchain-based platform for processing financial transactions to enhance secured and quick payments in the agriculture sector |
| | Pranto et al. [47] | Integrating with other technologies Enhancing fast payments | Development of a permissioned blockchain model for smart agriculture where transactions are transparent by integrating with other technologies and enhancing quick financial transactions |
| Blockchain for sustainability | Luzzani et al. [11] | Improving sustainable supply chains | Development of a blockchain-based sustainability certification system for the wine industry |
| | Song et al. [13] | Integrating with other technologies Improving sustainable supply chains | Design and development of a blockchain-based double chain structure for better sustainability by integrating with other technologies |
| | Dey et al. [10] | Integrating with other technologies Improving agricultural value chains | Development of a blockchain-based architecture for agriculture sustainability and improving agricultural value chains using the RAFT consensus algorithm |
| | Saurabh et al. [12] | Improving efficiency in supply chains Integrating with other technologies | A supply chain management information system using blockchain technology for improving efficiency in supply chains by integrating with other technologies |

**Table 4.** *Cont.*

| Themes | Author (s) | Enablers | Key Findings |
|---|---|---|---|
| Blockchain for Smart Agriculture | Bai et al. [48] | Enhancing trust among Stakeholders Enhancing ICT infrastructure | Development of a trust management system using blockchain technology to enhance trust among stakeholders by enhancing technology infrastructure |
| | Chen et al. [49] | Enhancing agricultural democratization | Development of a democratization system using blockchain that is integrated with IoT and other e-agriculture systems |
| | Hang et al. [50] | Enhancing efficiency and useability of data | Development of a fish farm management system using permissioned blockchain on Hyperledger Fabric for improved data useability |
| Blockchain for data security | Wu et al. [51] | Enhancing data security | Development of a security system using blockchain technology that protects the data collected from IoT devices and enhances security and guards against cyber attacks |
| | Anand et al. [52] | Integrating with other technologies Enhancing data security | Development of a blockchain-based framework that enhances data security in precision agriculture by integrating with other technologies such as IoT |

**Table 5.** Barriers of blockchain technology adoption.

| Themes | Author (s) | Barriers | Key Findings |
|---|---|---|---|
| Barriers of blockchain adoption | Yadav et al. [53] | Lack of government regulations Lack of trust among stakeholders | Emphasize the barriers such as government regulations, security and privacy concerns, regulatory uncertainties, and trust among the stakeholders in the agriculture industry |
| | Chen et al. [55] | Ease of use Lack of global standards | Highlight the barriers of the adoption of blockchain technology as the complexity of integration, the need for industrial cluster adoption, and a common and open data standard |
| | Kochupillai et al. [54] | Legal and ethical barriers | Highlight the barriers of adoption of blockchain such as legal and ethical issues |
| | Etemadi et al. [56] | Lack of government regulations Security and privacy concerns Scalability issues | The importance of technology maturity, clear regulatory provisions, scalability and bandwidth issues, and smart contract issues are identified as barriers |
| | Nurgazina et al. [57] | Security and privacy concerns Lack of global standards Resource capital requirements Lack of government regulations | Challenges in scalability, security, and privacy are identified as technical barriers for the adoption |
| | Dede et al. [58] | Lack of government regulations Security and privacy concerns Ease of use | Issues with the uncertain regulatory environment, scalability, technologically and socially understandable blockchain systems |

## 5. Discussion

*5.1. Enablers for Blockchain Technology Adoption in Smart and Sustainable Agriculture*

It can be seen that there is a significant development and application of blockchain technology for smart and sustainable agriculture. Even though there is an increase in scientific research contributions to the application of blockchain technology to agriculture in different contexts, the technology lacks full-pledged implementation.

There are various enablers that can contribute to the successful adoption of blockchain technology in smart and sustainable agriculture such as (a) stakeholder collaboration, (b) enhancing shared responsibilities of partners, (c) enhancing customer trust, quality of service, sustainable value chains, and infrastructure, (d) data security and useability, (e) improving efficiency in supply chains, and (f) enhancing agricultural democratization. The Ethereum blockchain network system brings all the stakeholders involved in agribusiness together to carry out robust transactions. Farmers can publish the agricultural assets as bids using smart contracts and customers can purchase them. According to Astill et al. [38], blockchain systems can enhance trust among stakeholders and improve transparency in agricultural supply chains. Blockchain technology makes it possible to improve trust and transparency among the participants in agricultural supply chains. Our findings also present an instance of shared responsibilities of partners as a consortium blockchain-based trustworthy system for agricultural supply chains. This system addresses the trading aspects where farmers and buyers can join a blockchain for trading. The consortium blockchain establishes trust and quality of goods and services on a reliable, transparent, trustworthy platform. It uses a permissioned consortium blockchain where the anonymity of buyers and farmers is maintained, and trading is managed via smart contracts. Blockchain technology has the potential to promote more sustainable agriculture supply chains. Smart contracts are used for interactions among the stakeholders. Another instance of a functional blockchain-based traceability system developed by Casino et al. [39] manages the dairy supply chains on a blockchain network. Blockchain can be integrated with other technologies such as IoT and AI. The BeefLedger system on a commercial blockchain-enabled provenance platform indicates the importance of integrating both blockchain and IoT for developing smart applications in precision agriculture. This system showcases blockchain technology capabilities which proves that it enhances the security, trust, traceability, and trustworthiness of data on a network. Blockchain technology indeed is a solution for most of the traceability, provenance, and performance challenges when integrated with IoT. Blockchain can be used to implement an intelligent decision-making system by integrating with AI. Such a system allows all the stakeholders of the blockchain to evaluate the quality of service, delivery of assets, punctuality, and data accuracy.

One of the traits of blockchain technology is the transparency of transactions on a network. An attempt to modify the data will be stored as a transaction on the network and all participants will know about the transaction. This will reduce tampering with data and fraud. A farm management system for food safety and traceability by using blockchain-based architecture, protocol stack and IoT is another instance that collects data about animals, employees' health, and packaging conditions and stores it in encrypted form on the blockchain network. The data is secured, and all participants can access the same data. HARA, a blockchain-based platform assists the agricultural industry and financial institutions to store, and processing secured transactions. In this instance, the application is accessible to farmers via a mobile phone. It facilitates the integration of various datasets in the agricultural industry such as land ownership and crop information and offers more affordable data management to participants. This platform connects farmers, suppliers, and financial institutions and thus, reduces the intermediaries and encourages a trustworthy environment.

We find that there are applications developed on smart and sustainable agriculture to manage consumer issues which explains the scope of adoption. We observe that the blockchain applications are divergent according to agricultural process lines, underlying platforms, and models. We identify that blockchain systems are tested for the applicability

of the technology in various areas of the agriculture domain. However, these systems do not address full-stack development for smart agriculture and sustainable agricultural practices. These systems are developed and tested for specific issues in agribusinesses to check the viability of blockchain technology adoption successfully.

It is evident from the review that blockchain technology is suitable for transparency, traceability, quick financial transactions, and data security in smart agriculture and food supply chains. There are real-time applications of blockchain technology for managing supply chains such as Pagonis Dairy, FarMarket and BeefLedger. These applications address the challenges and drawbacks of centralized systems and emphasize developing permissioned or private blockchains, or consortium blockchain ecosystems. These examples strongly recommend the adoption of blockchain technology for smart and sustainable agricultural practices. Blockchain has a major role in smart and sustainable agriculture where it can manage data acquisition, secure transaction processing, and better visibility of information. Blockchain and IoT-based technologies when integrated into smart agriculture will promote a great degree of transparency. The integrated system developed by using IoT and blockchain is a reliable solution for transparent food supply chains where IoT sensor devices, and RFID are used for data acquisition and blockchain for managing the IoT-generated data. Since blockchains are immutable, and cannot be tampered with, industry associations can rely on this information for sustainability certification. A system is developed to manage the certification process using blockchain technology to prove its corroboration as an adoption enabler. A blockchain-based Grape Wine Supply Chains Management Information System (GWSMIS) using smart contracts, machine learning, and cloud computing, which fosters disintermediation, democratic governance, and trust is developed by Luzzani et al. [11]. This is another classic example of blockchain application for sustainability in agriculture. Agricultural democratization can be achieved by combining existing agricultural systems with blockchain technology. Decentralization, transaction-sharing, and security make blockchain a general-purpose technology that has pervasiveness. When integrated with e-agriculture systems, blockchain can achieve digital agricultural democratization. FAIR data (Findability, Accessibility, Interoperability, Reusability) principles can be used for data integrity, and provenance, resulting in e-agriculture sustainability [60]. In this instance, the RAFT consensus algorithm by Dey et al. [10] for a permission-based blockchain system proves that blockchain makes the e-agriculture system efficient, decentralized, scalable, fault-tolerant, and interoperable. The smart agriculture model developed by Pranto et al. [47] is a permissioned blockchain system where the collected data resides on the MQTT server. In this model, smart contracts manage the role of each actor on the network from the pre-harvest period to the post-harvest period. This system stores data related to soil and water conditions, treatment with fertilizers, pesticides, waste reduction, labor, and human rights. All transactions are recorded on the blockchain for data transparency and traceability. This is another example of an adoption enabler that proves that blockchain connects all stakeholders to create a transparent environment for smart agriculture.

Another instance of an adoption enabler is the development of a permission-based blockchain application for fish farm management using Hyperledger Fabric [50]. Farmers can act as peers (nodes) to the blockchain and access information in a secure manner. Trust, data security, integrity, and data analysis are the major concerns for e-agriculture systems. Data collected from IoT sensors are always prone to security threats. A green supply chain framework by Bai et al. [48] to manage data security and integrity is a trust management system that records trust values on a blockchain. Here blockchain technology is employed to protect the access rights of a user, secure encrypted data, and avoid DDOS attacks from hackers. A network security system developed by Wu et al. [51] for protecting the data collected from IoT sensor devices and storing the data securely in a private blockchain is an example of another adoption enabler i.e., enhancing data security and useability.

We find that several authors address the drawbacks of an e-agricultural system for sustainability and suggest novel frameworks such as the double chain framework [13] along with a consensus method for better sustainability in agriculture. The RAFT consensus

algorithm [10] and the supply chains management information system [12] will benefit the agriculture industry by reducing the intermediaries, double spending and improving trust and democratic governance. In another case, blockchain has been used for precision agriculture to enhance the security of agricultural data. It is proven that the technology can be integrated with other technologies such as IoT for improved security of data and performance. One of the biggest challenges for smart and sustainable agriculture is the safety and security of agricultural data. Since the data is stored in encryption format on a blockchain and only accessed by authorized users, it is considered highly secure.

A few studies have developed blockchain-based applications and proven that the adoption of blockchain technology is possible in real-time [39–43,59]. These are the applications that make blockchain adoption possible and can be considered the early adopters of blockchain technology in agriculture. However, the adoption is limited to a few product lines to check the viability of blockchain technology. It can be observed that the adoption of blockchain technology is not integrated as a whole system for smart and sustainable agricultural practices. A blockchain-based system can only solve a few problems of smart agriculture management but not all. There are other processes where the technology needs to be studied further for adoption such as during the pre-harvesting and post-harvesting stages. The results suggest that although several companies have implemented blockchain solutions, further improvement of adoption processes and issues that cause delays in adoption are required to optimize smart and sustainable agriculture management. A homogenous characteristic is noticed among all blockchain applications testing this technology's viability. The final takeaways to address the research gap from the results are: (a) blockchain can be integrated with other emerging technologies, (b) it is compatible with the expansion and integration of existing systems, (c) requires technical knowledge and support for systems management, (d) requires organizational support and collaboration of stakeholders, and (e) requires upgradation of IT infrastructure. Therefore, a thorough study of the technological enablers and solutions to technical issues can help in building applications to advance sustainable agricultural practices.

### 5.2. Barriers for Blockchain Technology Adoption in Smart and Sustainable Agriculture

There are several barriers that need to be considered for blockchain technology adoption in smart and sustainable agriculture. These barriers are (a) the lack of government regulations, (b) resource capital requirements, (c) security and privacy concerns, (d) lack of standards, (e) trust, (f) scalability issues, (g) awareness, and (h) ease of use.

The lack of government regulations and regulatory uncertainty, resource capital requirements, security and privacy concerns, lack of standards, trust, scalability issues, awareness, and ease of use are barriers to blockchain technology adoption. In order to meet the increased food concerns of consumers, agrifood industries are forced to incorporate various technologies in their supply chains. In such circumstances, blockchain-enabled systems are playing a pivotal role. However, the adoption rate of blockchain in agriculture supply chains is slow due to the lack of appropriate government regulations and trust among agribusiness stakeholders. Blockchain adoption to smart and sustainable agriculture has both benefits and implementation barriers as the transparency of information on a blockchain-based system may lead to a lack of privacy for the stakeholders. Another threat is that there is a possibility of all stakeholders facing great challenges to define an open and standard data format in food supply chains. In addition to these barriers, the high resource capital requirements for implementing a blockchain system is another. However, organizations can have truly wide adoption of blockchain technology through collaboration and strong value proposition which requires government intervention and industry-level participation.

No doubt that blockchain technology offers a spectrum of benefits to the agriculture sector when integrated with other technologies such as IoT and AI. However, the adoption has not expanded to all aspects of smart and sustainable agriculture. Reasons for low expansion and slow adoption are the issues such as technical barriers to smart agriculture

and regulations and standards for sustainable agricultural practices. The selected articles were perceived as sufficient to provide interesting remarks. An in-depth study on the possibilities of blockchain adoption in agrobiodiversity is conducted to review the regulatory challenges and how this disruptive technology provides a solution to those barriers. The systematic review identifies a lack of trust among the stakeholders in the agrifood industry, a lack of traceability options, and expensive transactional costs. Blockchain technology provides a solution to address the issues of fraud, regulatory standards, and trust. However, there are certain barriers to adopting the technology because it will create new legal and ethical issues due to a lack of standards and protocols. New policies must be developed and implemented to use a blockchain-based solution to design a concrete incentive mechanism for transactions. Ethical issues are linked to the trustworthiness and integrity of codes and privacy concerns must be addressed at the early stage of development. The major concern would be the design and use of smart contracts towards fair coding and taking all stakeholders' interests equitably into consideration and free from bias. In this scenario, implementing and administering global-level standards are impractical. Barriers such as lack of confidentiality, lack of technological maturity, and users' credential loss are a few of the highly ranked challenges. Privacy and security, the suitability of blockchain, and high energy consumption are also considered potential barriers to adoption.

The adoption barriers can be sub-classified as scalability, security, and privacy challenges. There are enumerable disadvantages of the current blockchain applications in terms of scalability and security. However, some technical and policy-based resolutions to these challenges also exist. Technological immaturity, national and international regulations and standards, high costs of infrastructure and implementation, and substantial energy and power consumption are some of the major barriers to adoption. Moreover, the barriers to blockchain adoption are uncertain due to divergent laws and regulations, scalability and data verification process times, knowledge of technology, cyber-attacks, and the unrealistic thinking of blockchain as a 'one-size-fits-all'. Though the identified list of challenges is very short, the barriers are very significant and realistic. There are recommendations for developing novel consensus algorithms for data security, reliability, and trustworthiness at the data entry level and for updating or creating new policies for privacy by the regulatory authorities.

Table 5 summarizes the barriers to the adoption of blockchain technology. It is evident that there are significant barriers to adopting blockchain technology for smart and sustainable agriculture. However, these barriers will not make this technology obsolete. Researchers make several recommendations to manage the barriers to adoption. Chen et al. [55] state that the adoption of blockchain for any organization is like a double-edged sword. There are benefits as well as barriers to implementing a blockchain solution. The implementation is not just one technology but a suite of technologies that organizations must understand before making a decision on implementation. The technology is not a fit-for-all type but has tremendous characteristics and features that could solve business and operational problems. Blockchain is a continuously evolving technology and can be integrated successfully with other technologies such as IoT and AI if planned carefully. At a macro level, there are no barriers to adopting blockchain technology in the agriculture industry. However, a few technical and legal barriers and a lack of standards emerge due to the number of organizations and parties involved in the supply chain process and poor government regulations. A thorough implementation plan is needed for adopting this technology for smart and sustainable agriculture.

*5.3. Recommendations for Blockchain Technology Adoption in the Agriculture Sector*

Blockchain technology could be used to supplement agrifood traceability, quick and reliable payments, transparency, security, and integrity of agricultural supply chains [17]. There are sufficient cases that prove that blockchain systems can reduce the intermediaries and improve trustworthiness among the stakeholders in agriculture supply chains. Many blockchain applications are available that address the issues of the agriculture sector such

as transparency, data security and useability, traceability, information asymmetry, and provenance. In this study, we find that blockchain technology can address the issues of smart agriculture and sustainable agricultural practices. However, blockchain technology needs to be integrated with the existing IoT-based systems and legacy systems to achieve its complete potential. The technical barriers such as availability and management of infrastructure, scalability, interoperability, and high-power consumption limit its adoption. However, blockchain has the capacity to be used for secured machine-to-machine communication among IoT devices, financial transactions, Government-to-Business and Business-to-Business transactions for smart and sustainable agriculture even at a large scale. As with more BaaS (Blockchain as a Service) providers and distributed ledgers emerging, there will be more reliable blockchain networks, and applications developed that will make a vast majority of agribusinesses adopt this technology. It is necessary to upgrade policies and industry-wide process standardization, government regulations, global standards, and best practices before a successful technology adoption.

### 5.3.1. Interoperability

This study reveals that blockchain technology needs to be integrated with IoT and AI for smart and sustainable agriculture. The integration of various techniques, such as machine learning, into blockchain is suggested for sustainable practices. For Agriculture 4.0 and green technologies, smart contract-enabled payment systems (digital currencies), IoT sensors, GPS, GIS, mapping tools, and wireless equipment are very critical in order to make appropriate decisions and mitigate information asymmetry issues and adverse selection of value chain intermediaries. Further research is needed towards the interoperability of these emerging technologies. Research on innovative systems reveals that cooperation between organizations and universities can boost innovation and sustainable development [6]. Integration of blockchain and other technologies, such as IoT and machine learning, needs to be developed beyond food supply chains and payment mechanisms in agriculture.

### 5.3.2. Best Practices and Regulations

The absence of global uniform standards and best practices with which all the participants can comply is one of the biggest barriers to blockchain technology adoption in the agriculture industry. These standards will improve market stability, bring ethical behavior to participants, and regulate fraudulent transactions, increasing public trust and technology adoption. Disintermediation, one of the barriers, will allow direct financial transactions that lead to money laundering and fraud. There is a need for governance and centralized authority for surveillance. The involvement of industry chains and government regulatory authorities is very important towards the development of best practices, policies, and regulations. Further research is needed to consider the role of governments and industrial experts in enforcing the regulations at different levels of agricultural supply chains.

### 5.3.3. Security and Privacy

The lack of data security and privacy has an adverse effect on the accuracy of agricultural transactions. Farmers and other intermediaries on a blockchain-based system are apprehensive about the transactions and their credentials. Moreover, the members of the food supply chain may not be willing to share their data with the competitors as they may think that the blockchain system is too transparent. Therefore, more research needs to be done to decide on the role and function of governments on this technology to implement policies and procedures for data protection and control users' behavior as well as reducing ownership and reservation of data exchange.

### 5.3.4. BaaS and DLT Providers

For data management, a secured and trusted blockchain distributed ledger is critical to smart and sustainable agriculture that can improvise real-time decision making and integrate local agrifood networks into global value chains. A typical blockchain-based smart

agriculture system costs very high in terms of development, consumption of electricity and maintenance [10]. There are blockchain service providers and several distributed ledgers available that provide technical support to manage the transactions on blockchain such as Oracle and Microsoft. However, more research and development are needed for a comprehensive suite of blockchain applications or a smart agriculture system that supports pre-harvest and post-harvest conditions. Farmers and food suppliers can reduce the costs of operation and avoid technical barriers by using BaaS services.

In summary, we can see that there are potential benefits of blockchain technology adoption for the agriculture sector. To ensure that blockchain technology is successfully adopted, it is critical to understand the main barriers and enablers influencing its adoption. Table 6 summarizes the barriers, enablers and recommendations for blockchain technology adoption in smart and sustainable agriculture.

**Table 6.** Barriers, enablers and recommendations for blockchain technology adoption in smart and sustainable agriculture.

| Barriers | Enablers | Recommendations |
|---|---|---|
| - Lack of government regulations<br>- Lack of global standards<br>- Legal and ethical barriers<br>- Lack of trust among stakeholders | - Stakeholder collaboration<br>- Enhancing shared responsibilities of partners<br>- Enhancing customer value, quality of service and trust<br>- Improving efficient sustainable supply chains and value chains | - Best practices, industry-wide standards, and regulations |
| - Security and privacy concerns | - Enhancing ICT infrastructure, audit, and fast payments<br>- Enhancing agricultural democratization, data useability and security | - Security and privacy protocols and global-level coding standards |
| - Scalability issues<br>- Ease of use<br>- Resource capital requirements | - Integrating with other technologies<br>- Increasing product quality via traceability | - Interoperability with other emerging technologies<br>- BaaS and DLT providers |

## 6. Conclusions

Several studies have shown that blockchain technology together with sustainable practices can help achieve smart and sustainable agriculture. Thus, this systematic review explores the barriers and enablers of blockchain adoption for smart and sustainable agriculture. The PRISMA technique is adopted for data collection, out of which 26 articles are relevant toward the application of blockchain technology in the agriculture sector. We have identified 18 blockchain adoption enablers and 8 major barriers from the review process. The blockchain adoption enablers include enhancement in stakeholder collaboration, shared responsibilities, customer value, quality of service, trust, sustainable supply chains and value chains, democratization, and data useability. There are, however, several barriers to a slow blockchain adoption rate due to unclear government regulations, technology immaturity, scalability and bandwidth issues, policies on privacy, information sharing, power consumption, high establishment costs, and security. It is also observed that the technical barriers to blockchain adoption can be easily resolved with blockchain service providers' collaboration and support. However, defining global standards, government regulations and centralized authorities' involvement may take more time to resolve, and this causes more delays in blockchain adoption.

Our study has some limitations that could lead to future research opportunities in addition to the ones identified in the Discussion section. The first limitation is related to the keywords chosen and the topic selected. Future research needs to consider the inclusion of more databases, other technologies associated with blockchain, and various blockchain-based applications. Nonetheless, this will open the door to new exploratory

studies for a better understanding of how blockchain can shape the future, leading to a more sustainable agrifood sector.

**Author Contributions:** Conceptualization, G.K.A. and S.W.; methodology, G.K.A.; validation, S.W. and S.G.; formal analysis, G.K.A.; writing—original draft preparation, G.K.A.; writing—review and editing, S.W., S.G. and S.M.; supervision, S.W., S.G. and S.M.; project administration, G.K.A. All authors have read and agreed to the published version of the manuscript.

**Funding:** This research received no external funding.

**Data Availability Statement:** Not applicable.

**Conflicts of Interest:** The authors declare no conflict of interest.

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
