# Peer review of "A Systematic Review of Blockchain Technology Adoption Barriers and Enablers for Smart and Sustainable Agriculture"

_2504-2289, doi:10.3390/bdcc7020086_

Round 1
Reviewer 1 Report
I consider that the article is well written, but I have some recommendations:
In Abstract is necessary to emphasize better the aim of this study.
The Literature review on the subject is not relevant, and the Research Questions are not motivated and are not the result of a "gap" in the literature.
The criteria of exclusion are not clearly explained.
In the model of research is not explained the necessity/motivation to include a bibliometric analysis.
I consider that it is necessary to extend the conclusion section with respect to the structures recommended in the matter.
Over 90% of references in-text are placed at the end of the sentence. Please, avoid this pattern, if possible.
1. What is the main question addressed by the research?
I believe that for me is not relevant to present the main question, but is important to evaluate the existence of this in the appropriate section.
As you can see in the Introduction section, there are a set of 3 research questions. So, there is not a main question of research.
"RQ1. What is the role of blockchain technology for smart and sustainable agriculture?
RQ2. What are barriers and enablers in blockchain technology adoption for smart 83 and sustainable agriculture?
RQ3. What are the future trends for blockchain technology adoption in the agriculture sector?"
Only, one question (RQ2) is in accord with the title.
In this context, it's necessary to update this paper for one principal question of research or it is necessary to change the title and formulate a general research question with 3 subsequent research questions.
2. Do you consider the topic original or relevant in the field? Does it address a specific gap in the field?
From my point of view, I consider relevant and original the topic as a subject of research, but is not very clearly presented (with relevant references) the specific gap in the field.
So, it is necessary to motivate in a clear manner the gap in the field with relevant references.
3. What does it add to the subject area compared with other published material?
As you can see, in this paper is missed the section on Literature review, and is not presented in a clear manner if there are previous works on the subject of SLR in the field emphasised in the title.
We can't identify in this article a discussion about similar SLR in the matter.
So, it is very important to identify a similar paper on SLR in the field of the title, and motivate and adapt the research parameters of this article in accord with these aspects.
4. What specific improvements should the authors consider regarding the methodology? What further controls should be considered?
Regarding methodology, it is not clear why it is necessary this superficial "Bibliometric analysis" from section 2.3, and also the "Content analysis" from section 2.4. Regarding the key words of search and the query sentence is not motivated why the keywords are included in the searching process. Also, is not presented the string of searching for a good understanding of the logic of research and ensuring the replicability of the research.
In my opinion, the both sections must be presented in the Research methodology as a part of this, in a kind of research model, and in the Data and results section must be presented the application of this methodology. As you can see, there is a section with results, but between sections research method and results is included the section no 3 a kind of mix between theory and results (with the role of literature review).
So, it is necessary to clarify and explain the full research methodology starting from the PRISMA model with a clear separation from the results. The results must be presented in a clear manner, also. The literature review aspects must be discussed before the Research methodology.
5. Are the conclusions consistent with the evidence and arguments presented and do they address the main question posed?
In the Conclusion(s) section there are presented principal ideas about the research and the results of the research, but in a general manner without a
presentation on the three research questions assumed in the research.
I consider that is necessary that the authors to reconsider the conclusion, in accordance with the appropriate research questions which are established in the Introductions and Literature review.
6. Are the references appropriate?
Yes, I consider that the references are appropriate, but of course, it is possible to add as a comment that will be good for authors to analyse the Prisma statements, and also other articles published in the field of SLR on MDPI in open access.
7. Please include any additional comments on the tables and figures.
It is necessary to present the source of the Tables and Figures.
Author Response
Dear Reviewer,
Thank you for reviewing our paper. Please find the attached document for detailed responses.
Best regards,

Reviewer 2 Report
The topic is very interesting and I hope that the authors will consider my comments.
“These studies state that there is a need for reliable and trustless” à Do you mean “trustful”?
“The results provide recommendations for blockchain adoption” à The results should also include “Barriers and Enablers”, not only recommendations. When it is written in this way, then the connection with the topic (title) is somehow lost.
Why does the title of the paper contain only RQ2?
Sometimes it seems as if the text is repeated in some parts, for example the text in the introduction and the chapter “3. Blockchain for agriculture”. I believe that the theoretical description of Blockchain and agriculture supply chains would belong to a kind of "Background" or “Theoretical framework”, which should be placed before the Methodology.
“A significant increase can be seen from 2019 in the number of articles published on blockchain technology” à do you mean in general, or the articles related to agriculture? Later, "finance industry" is mentioned.
“it can be seen that the trend is moving towards agriculture, machine learning and IoT networks in 2020.” à Where exactly can it be seen? Besides, does not figure 3 belong in the Results?
Sometimes, I am not sure what is the difference between chapter 3 and the "Results" chapter. Some results are even presented in the methodology (for example: “In the content analysis, six themes are identified in the selected articles…”).The methodology should contain the methodology, not the results.
Although I really see that a lot of effort has been put in, a lack of the focus is visible (despite the emphasized research questions etc.). I also recommend the authors to focus on the topic (title) of the manuscript.
Author Response

(The authors gave the same response as above.)

Round 2
Reviewer 2 Report
To begin, the manuscript is much better. However, I recommend that you also adopt the following comments:
Abstract:
“The results provide adoption enablers, and barriers, for blockchain adoption in agrifood sector such as (1) stakeholder collaboration, (2) enhance customer trust, (3) democratization, and (4) global standards, industry level best practices and policies.” à The barriers are definitely not collaboration, enhanced trust etc. The text must make sense.
“Many studies [13-16] have been conducted on understanding the effect of blockchain for smart and sustainable agriculture. These studies showed that blockchain technology can provide a solution to the issues relating to double spending, information asymmetry, traceability, and data security in agriculture sector [25,36,37]” à Authors first mentioned many studies [13-16] studies (only 3), then continued with "these studies", but this time, other references are mentioned. It is not logical.
“Based on the review, the results show that there are several barriers that need to be considered for blockchain technology adoption in smart and sustainable agriculture such as:” à results of your research are mentioned in Introduction. It is necessary to have a clear understanding of what each chapter should contain and what the goal of a particular chapter is. What is the purpose of the Results chapter then?
In Table 1, Inclusion is mentioned twice. It is not understandable in this form
Lines 307-333à It is very difficult to follow. It would be better if you could shape the sentences better and separate them into meaningful paragraphs. In addition, this text is all already in the table, so I am not sure that everything needs to be written again (unnecessary repetition). It is certainly good to describe in the text what is in the table, but then add some of your thinking, connect a little, and not just list what is already in the table.
Author Response
Dear Reviewer,
Thank you for your constructive feedback. We have made changes to our manuscript and also attached our point-by-point response.
Best regards,
Gopi Akella
